# Just rephrase it! Uncertainty estimation in closed-source language models via multiple rephrased queries

## Abstract

We explore estimating the uncertainty of closed-source LLMs via multiple rephrasings of an original base query. Specifically, we ask the model, multiple rephrased questions, and use the similarity of the answers as an estimate of uncertainty. We diverge from previous work in i) providing rules for rephrasing that are simple to memorize and use in practice ii) proposing a theoretical framework for why multiple rephrased queries obtain calibrated uncertainty estimates. Our method demonstrates significant improvements in the calibration of uncertainty estimates compared to the baseline and provides intuition as to how query strategies should be designed for optimal test calibration.

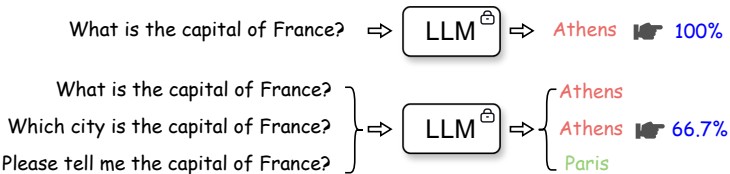

Figure 1: **Multiple rephrased queries for uncertainty estimation.** Querying a closed-source LLM only once with a base query may yield an incorrect top-1 prediction with $100\%$ confidence to this singular prediction. Querying the model multiple times with rephrased versions of the base query produces different answers equivalent to $66.6\%$ confidence.

## 1 Introduction

Closed-source LLMs are prone to generating highly convincing but false information, a problem known as "hallucinating" (Huang et al., 2023; Ji et al., 2023). It is folk wisdom that one approach for estimating LLM uncertainty, even with such limited access to the model, is to query it multiple times (Wang et al., 2022; Xiong et al., 2023). This approach is based on the premise that LLM-generated text is frequently stochastic by design, as the next generated token is chosen through nucleus sampling (Holtzman et al., 2019) or top-k decoding (Fan et al., 2018; Radford et al., 2019). Wang et al. (2022) and Xiong et al. (2023) proposed to use the consistency of multiple answers as an estimate of uncertainty. Xiong et al. (2023) furthermore proposed to add "noise" to the base query at each repetition, through misleading hints.

In this work, we delve deeply in, refine, and theoretically analyze multiple queries for uncertainty estimation. Given a base query, we restrict ourselves to submitting rephrased versions of the base

Submitted to 38th Conference on Neural Information Processing Systems (NeurIPS 2024). Do not distribute.

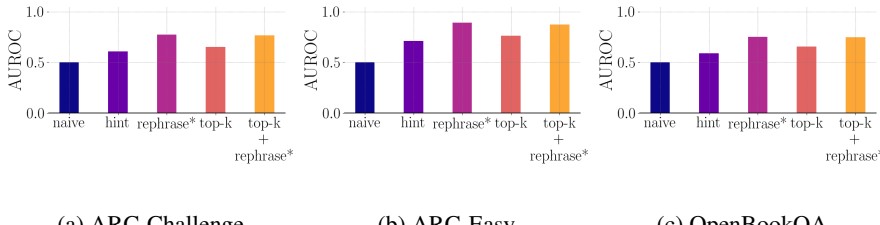

| (a) ARC-Challenge | (b) ARC-Easy | (c) OpenBookQA |

Figure 2: We plot the AUROC averaged over all models for each dataset and for each uncertainty estimation method. We observe that top-k improves over the naive top-1 decoding. Furthermore, the best rephrasing method (denoted as rephrase*) improves the AUROC significantly in all cases.

query to an LLM, checking the consistency of the answers, and using the result as an estimate of uncertainty. Concretely our contributions are the following:

- We test four simple strategies for creating multiple rephrased queries, and find that they result in significant calibration gains over baselines.
- We propose a theoretical model for multiple rephrased queries on a simplified top-1 and top-k (Holtzman et al., 2019) decoding setting. Given multiple rephrased queries, our analysis shows that i) it is possible to recover the probability of the answer under the inaccessible categorical distribution of the LLM ii) top-k decoding then simply tempers our uncertainty. Crucially we show empirically that our uncertainty estimates are close to what could be obtained when having access to the last layer logits.

## 2 Rephrasing drastically improves calibration for top-1 decoding

Let $f : \mathcal{X} \to \mathcal{Y}$ be an LLM which takes $\boldsymbol{x}$ an input query in the form of a multiple choice question, and outputs $y$, an answer. *We first consider top-1 decoding such that the answers of the LLM are deterministic.* We consider randomized transformations of the base query $\mathcal{T}(\boldsymbol{x}) \sim \tau$ in the form of rephrasings of the query, and the most probable answer under the transformations $A = \operatorname{argmax}_i \mathbb{P}\left(f(\mathcal{T}(\boldsymbol{x})) = i\right)$. In a multiple choice question setting (which can be seen as a multi-class classification problem), we will use $A$ as the predicted class and

$$p_A(\boldsymbol{x}) = \mathbb{P}\left(f(\mathcal{T}(\boldsymbol{x})) = A\right),$$

as our confidence about this prediction (here the predicted class coincides with a predicted token denoting this class). We consider four types of rephrasings, with an increasing level of modification to the original query: (1) reword: replacing words with synonyms; (2) rephrase: modifies the structure of the original query; (3) paraphrase: reconstructs the original query; (4) expansion: elaborate the query. In general, we perform the rephrasings with a separate instance of the same model that responds to the queries. We estimate $p_A(\boldsymbol{x})$ using Monte Carlo sampling with 10 draws from $\mathcal{T}(\boldsymbol{x}) \sim \tau$ to estimate uncertainty with our method unless stated otherwise.

We used three different models, the Llama-2 7B model, the Llama-2 13B model (Touvron et al., 2023) and the Mistral 7B model (Jiang et al., 2023). We tested our framework on three multiple choice tasks: ARC-Challenge, ARC-Easy (Clark et al., 2018), and Openbookqa (Mihaylov et al., 2018). Following Kojima et al. (2022), we extract the answer from LLM-generated texts by looking at the first appearance of A/B/C/D. To test for calibration we used standard calibration metrics, including the ECE and TACE (Naeini et al., 2015), Brier score (Murphy, 1973) and AUROC (Murphy, 2012).

We plot the AUROC results of all methods averaged over all models for each dataset in Figure 2. We see that the best rephrasing method outperforms top-1 (naive) and top-k decoding as well as the hint based rephrasing approach. In Appendices E and D we show that we also match or outperform Chain of Thought (CoT) prompting and Temperature Sampling Wei et al. (2022).

## 3 Rephrasing works as well as having access to the last layer logits

We now derive a proposition that elucidates why $p_A(\boldsymbol{x})$ results in calibrated estimates of uncertainty.

Table 1: Comparisons between our rephrasing methods and white-box logit uncertainty estimation. We see that our rephrasing methods achieve similar calibration to what would be achieved if we had access to last layer logits. This is evident both in the AUROC and TACE as well as the Brier score, which also accounts for accuracy.

| Dataset | Model | Method | Acc ↑ | ECE ↓ | TACE ↓ | Brier ↓ | AUROC ↑ |
|---|---|---|---|---|---|---|---|
| ARC-C | Mistral-7B | logits | 0.742 | 0.252 | 0.075 | 0.503 | 0.741 |
| | | expansion | 0.602 | 0.133 | 0.099 | 0.509 | 0.847 |
| | Llama-2-7B | logits | 0.483 | 0.362 | 0.168 | 0.853 | 0.621 |
| | | expansion | 0.373 | 0.112 | 0.153 | 0.778 | 0.687 |
| | Llama-2-13B | logits | 0.508 | 0.132 | 0.141 | 0.704 | 0.669 |
| | | reword | 0.445 | 0.084 | 0.119 | 0.714 | 0.721 |
| ARC-E | Mistral-7B | logits | 0.866 | 0.128 | 0.037 | 0.264 | 0.818 |
| | | reword | 0.753 | 0.045 | 0.062 | 0.297 | 0.931 |
| | Llama-2-7B | logits | 0.672 | 0.190 | 0.098 | 0.493 | 0.779 |
| | | rephrase | 0.535 | 0.131 | 0.117 | 0.603 | 0.830 |
| | Llama-2-13B | logits | 0.617 | 0.060 | 0.094 | 0.498 | 0.763 |
| | | expansion | 0.524 | 0.078 | 0.12 | 0.552 | 0.893 |
| OBQA | Mistral-7B | logits | 0.655 | 0.298 | 0.085 | 0.602 | 0.705 |
| | | reword | 0.552 | 0.105 | 0.102 | 0.592 | 0.796 |
| | Llama-2-7B | logits | 0.478 | 0.277 | 0.147 | 0.758 | 0.642 |
| | | expansion | 0.362 | 0.083 | 0.138 | 0.775 | 0.678 |
| | Llama-2-13B | logits | 0.418 | 0.168 | 0.135 | 0.723 | 0.650 |
| | | rephrase | 0.428 | 0.095 | 0.14 | 0.729 | 0.73 |

**Proposition 3.1.** *Let $f : \mathcal{X} \to \mathcal{Y}$ be an LLM, $\boldsymbol{x}$ is a base query and $\mathcal{T}(\boldsymbol{x}) \sim \tau$ is some randomized transformation of the base query. Let*

$$p_A(\boldsymbol{x}) = \mathbb{P}\left(f(\mathcal{T}(\boldsymbol{x})) = A\right), \tag{1}$$

*be the probability of sampling the most probable answer $A \in \mathcal{Y}$ under transformations $\mathcal{T}(\boldsymbol{x}) \sim \tau$. Let $\boldsymbol{z}_{mean} + \epsilon_{rephrase}$ be the latent representation of $\boldsymbol{x}$ under $\mathcal{T}(\boldsymbol{x})$ at the final LLM layer, where $\boldsymbol{z}_{mean}$ is the mean representation and $\epsilon_{rephrase}$ is some additive noise. Let $\mathbf{w}$ be the separating hyperplane between the most probable answer $A$ and the second most probable answer $B$. Assuming that $\mathbf{w}^\top \epsilon_{rephrase} \sim \rho$ follows a logistic distribution with $\mu = 0$ and $s = 1$ then*

$$p_A(\boldsymbol{x}) = p(A|\boldsymbol{z}_{mean}, f) \tag{2}$$

*where $p(A|\boldsymbol{z}_{mean}, f)$ is the probability of $A$ given $\boldsymbol{z}_{mean}$ under the categorical distribution of the final layer.*

We prove the above for the binary case of two classes $A$ and $B$ in Appendix C, but expect that it should be sufficiently informative in multi-class settings when $A, B$ are much more probable than other classes. A crucial assumption for recovering well-calibrated predictions is that $\mathbf{w}^\top \epsilon_{rephrase} \sim \rho$ follows a logistic distribution with $\mu = 0$ and $s = 1$. We test this assumption by computing the cumulative of $\rho$ for our different experimental setups. In Figure 3c we find and plot the empirical cumulative using a Kolmogorov-Smirnov test (Smirnov, 1948) and $S = 100$ MC samples of $\rho$ for Mistral-7B, ARC-Challenge, and the "expansion" rephrasing method. We see that the indeed the cumulative is approximately logistical validating our prediction (the confidence bands cover different queries $\boldsymbol{x}$). In Table 1 we use the logits of the answers as an oracle white-box uncertainty estimate. Specifically, we apply the softmax function and use the probability of the most probable class as our estimate of uncertainty. We compare the results of this method with the best rephrasing method (in terms of Brier). We observe that our uncertainty estimates that are similar to what we would get if we had access to the last layer logits.

## 4 For top-k decoding, rephrasing tempers predictive uncertainty

In practice, the assumptions of the above proposition are too restrictive. In particular, decoding in LLMs is performed with top-k decoding or nucleus sampling instead of top-1 decoding. Furthermore

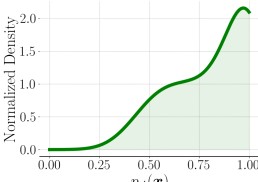 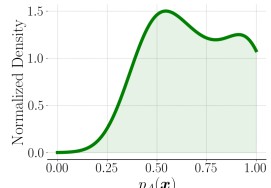 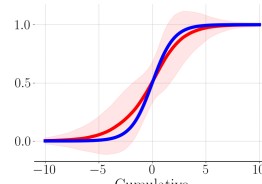

(a) $p_A(\boldsymbol{x})$ for top-k without rephrasing

(b) $p_A(\boldsymbol{x})$ for top-k with rephrasing

(c) Logistic (blue), and empirical cdf (red)

Figure 3: We plot the distribution of $p_A(\boldsymbol{x})$ for the case of top-k decoding with and without rephrasing, for all datasets, models, and rephrasing methods. We see that rephrasing primarily acts to temper the probability of the most probable class $A$, thus making the model less confident and possibly better calibrated. We also plot the logistic (blue), and empirical cdf (red) for $\mathbf{w}^{\top}\epsilon_{rephrase} \sim \rho$ for Mistral-7B, ARC-Challenge, and the "expansion" rephrasing method for top-1 decoding. $\rho$ is often close to a logistic distribution.

while for an oracle choice of the rephrasing intensity the modeling assumption that $\mathbf{w}^{\top}\epsilon_{\eta} \sim \rho$ follows a logistic distribution with $\mu = 0$ and $s = 1$ might be correct, in general, the variance of the noise in latent space is unknown. It is thus illustrative to consider an extension of our toy model. The following proposition explores these extensions.

**Proposition 4.1.** *Let $g : \mathbb{R}^{d_\eta} \to \mathcal{Y}$ be the final encoder layer of an LLM, $\boldsymbol{x}$ is a base query and $\mathcal{T}(\boldsymbol{x}) \sim \tau$ is some randomized transformation of the base query. Let*

$$p_A(\boldsymbol{x}) = \mathbb{P}\left(f(\mathcal{T}(\boldsymbol{x})) = A\right), \tag{3}$$

*be the probability of sampling the most probable answer $f(\boldsymbol{x}) = A \in \mathcal{Y}$ under transformations $\mathcal{T}(\boldsymbol{x}) \sim \tau$. Let $\boldsymbol{z}_{mean} + \epsilon_{topk} + \epsilon_{rephrase}$ be the latent representation of $\boldsymbol{x}$ under $\mathcal{T}(\boldsymbol{x})$ at the final LLM layer, where $\boldsymbol{z}_{mean}$ is the mean representation and $\epsilon_{topk}$ is additive noise resulting from the top-k decoding and $\epsilon_{rephrase}$ is additive noise resulting from the rephrasings $\mathcal{T}(\boldsymbol{x})$. Assuming that $\mathbf{w}^{\top}(\epsilon_{topk} + \epsilon_{rephrase}) \sim \rho$ approximately follows a logistic distribution with $\mu = 0$ and $s = \sqrt{s_{topk}^2 + s_{rephrase}^2}$ then*

$$p_A(\boldsymbol{x}) \approx 0.5 + \frac{1}{\sqrt{s_{topk}^2 + s_{rephrase}^2}}(p(A|\boldsymbol{z}_{mean}, f) - 0.5) \tag{4}$$

*where $p(A|\boldsymbol{z}_{mean}, f)$ is the probability of $A$ given $\boldsymbol{z}_{mean}$ under the categorical distribution of g.*

The approximation relies on linearizing the involved functions, however, it is illustrative of the effect of both $s_{topk}^2$ and $s_{rephrase}^2$. In particular, we see that both $s_{topk}^2$ and $s_{rephrase}^2$ act to *temper* the probability $p(A|\boldsymbol{z}_{mean}, f)$ under the categorical distribution of g. This highlights why using rephrasings with an appropriate temperature might improve the calibration in downstream tasks. In previous works, tempering of the categorical distribution has been found to significantly improve the calibration of deep neural networks (Guo et al., 2017).

Figure 3 shows that in accordance with proposition 4.1 rephrasing acts primarily to temper the probability of the top class. In out detailed results in Appendix E, this often improves calibration significantly in terms of ECE, and AUROC especially for smaller models.

# 5 Discussion

We conducted a thorough analysis of rephrased queries as a method for obtaining calibrated predictions from closed-source LLM models. Notably, we found that two simple methods; making the query more verbose, and substituting words with their synonyms, provide a straightforward means of identifying false positives. The appeal of our approach lies in its practicality, as it requires only basic language and arithmetic skills by the end user to obtain meaningful uncertainty estimates. Exciting future directions include learning optimal rephrasing rules in a data-driven manner, to be used in conjunction with a rephrasing LLM. While we tested on the multiple choice question setting for ease of evaluation, we expect our results to also hold for open-ended text generation.

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

Table 3: Different Prompts to Induce Consistency Confidence

| | Consistency Confidence Inducing Prompts |
|---|---|
| Hint1 | I think the answer should be |
| Hint2 | I read online the answer is |
| Hint3 | I vaguely remember the answer is |

## A  Prompt template

We present our prompt template for initiating rephrases with a one-shot example in Table 2. Note that we only present and rephrase questions without revealing choices, to reduce unnecessary bias to rephrases when presented with answer choices.

| Method | Prompt |
|---|---|
| reword | [INST]Reword the following question:
George wants to warm his hands quickly by rubbing them. Which skin surface will produce the most heat?
Respond with the reworded question only: [\INST]
George seeks to heat his hands swiftly by rubbing them. Which skin area will generate the maximum heat?
[INST]Reword the following question: {question}
Respond with the reworded question only: [\INST] |
| rephrase | [INST]Rephrase the following question:
George wants to warm his hands quickly by rubbing them. Which skin surface will produce the most heat?
Respond with the rephrased question only: [\INST]
What type of skin texture on George's hands would generate the most heat through rapid rubbing to warm them effectively?
[INST]Rephrase the following question: {question}
Respond with the rephrased question only: [\INST] |
| paraphrase | [INST]Semantically paraphrase the following question:
George wants to warm his hands quickly by rubbing them. Which skin surface will produce the most heat?
Respond with the semantically paraphrased question only: [\INST]
How can George induce the highest thermal output by briskly rubbing his hands, and which part of the skin would be most effective?
[INST]Semantically paraphrase the following question: {question}
Respond with the semantically paraphrased question only: [\INST] |
| expansion | [INST] Expand the following question with additional context:
George wants to warm his hands quickly by rubbing them. Which skin surface will produce the most heat?
Respond with the expanded question only: [\INST]
In the context of seeking immediate relief from the biting cold and understanding the mechanisms behind heat generation through friction, what type of skin texture on George's hands would most effectively generate heat by rapid rubbing?
[INST]Expand the following question with additional context: {question}
Respond with the expanded question only: [\INST] |

Table 2: Prompt templates for one-shot rephrasing, with rephrasing methods listed on the left and corresponding prompt on the right. The user instructions are colored in blue and surrounded by the instruction token, whereas model response demonstrations are colored in orange.

We followed the instructions in Xiong et al. (2023) to generate "hint" based rephrasings. Specifically, to generate a rephrased query given a base query, we appended one of the following three weak claims (as they found weak claims outperform other types of hints) together with a random class from the available ones.

| Method | Question |
| --- | --- |
| original | What part of the digestive system first causes chemical changes to food? A. Teeth in the mouth. B. Saliva in the mouth. C. Enzymes in the stomach. D. Enzymes in the small intestine. |
| reword | Which region of the gastrointestinal tract initiates the initial chemical modifications to food intake? A. Teeth in the mouth. B. Saliva in the mouth. C. Enzymes in the stomach. D. Enzymes in the small intestine. |
| rephrase | In what region of the digestive system does the food undergo its initial chemical transformations? A. Teeth in the mouth. B. Saliva in the mouth. C. Enzymes in the stomach. D. Enzymes in the small intestine. |
| paraphrase | At what point in the digestive process do initial chemical transformations of food occur and which section of the system carries out this function? A. Teeth in the mouth. B. Saliva in the mouth. C. Enzymes in the stomach. D. Enzymes in the small intestine. |
| expansion | Considering the intricate process by which our bodies break down and absorb nutrients from food, which specific organ or region within the digestive system initiates the essential biochemical transformations through enzyme secretion and the beginning of the digestion process? A. Teeth in the mouth. B. Saliva in the mouth. C. Enzymes in the stomach. D. Enzymes in the small intestine. |

Table 4: Rephrasing examples generated by Mistral-7B, with rephrasing methods listed on the left and corresponding rephrases on the right.

| Method | Question |
| --- | --- |
| original | What part of the digestive system first causes chemical changes to food? A. Teeth in the mouth. B. Saliva in the mouth. C. Enzymes in the stomach. D. Enzymes in the small intestine. |
| reword | What section of the digestive system initiates chemical alterations to sustenance? A. Teeth in the mouth. B. Saliva in the mouth. C. Enzymes in the stomach. D. Enzymes in the small intestine. |
| rephrase | Which portion of the digestive system initially catalyzes the biochemical transformation of ingested sustenance? A. Teeth in the mouth. B. Saliva in the mouth. C. Enzymes in the stomach. D. Enzymes in the small intestine. |
| paraphrase | Which digestive organ releases enzymes that initiate chemical breakdown within ingested sustenences, leading to nutrient extraction and energy release? A. Teeth in the mouth. B. Saliva in the mouth. C. Enzymes in the stomach. D. Enzymes in the small intestine. |
| expansion | In the context of the digestive process and the breakdown of nutrients, which portion of the digestive system initiates the chemical transformations that result in the nutrient absorption and energy production? A. Teeth in the mouth. B. Saliva in the mouth. C. Enzymes in the stomach. D. Enzymes in the small intestine. |

Table 5: Rephrasing examples generated by Llama2-7B, with rephrasing methods listed on the left and corresponding rephrases on the right.

# B  Rephrase generations

Here, we present additional generated rephrasings by Mistral-7B, Llama2-7B and Llama2-13B in Table 4, Table 5 and Table 6.

# C  Additional Proofs

**Proposition C.1.** *Let $f : \mathcal{X} \to \mathcal{Y}$ be an LLM, $\boldsymbol{x}$ is a base query and $\mathcal{T}(\boldsymbol{x}) \sim \tau$ is some randomized transformation of the base query. Let*

$$p_A(\boldsymbol{x}) = \mathbb{P}\left(f(\mathcal{T}(\boldsymbol{x})) = A\right),\tag{5}$$

| Method | Question |
|--------|----------|
| original | What part of the digestive system first causes chemical changes to food? A. Teeth in the mouth. B. Saliva in the mouth. C. Enzymes in the stomach. D. Enzymes in the small intestine. |
| reword | Which section of the gastrointestinal tract initiates the chemical transformation of sustenance? A. Teeth in the mouth. B. Saliva in the mouth. C. Enzymes in the stomach. D. Enzymes in the small intestine. |
| rephrase | In which section of the digestive system does the initial chemical breakdown of food occur? A. Teeth in the mouth. B. Saliva in the mouth. C. Enzymes in the stomach. D. Enzymes in the small intestine. |
| paraphrase | In the digestive process, where do crucial transformations initially occur to break down nutrients? A. Teeth in the mouth. B. Saliva in the mouth. C. Enzymes in the stomach. D. Enzymes in the small intestine. |
| expansion | Taking into account that human digestive system's several organs coordinate to breakdown, absorb, and expel waste, which part of the gastrointestinal system would have the most significant logic-based influence on the breakdown of food into usable components, prior to the nutrient absorption? A. Teeth in the mouth. B. Saliva in the mouth. C. Enzymes in the stomach. D. Enzymes in the small intestine. |

Table 6: Rephrasing examples generated by Llama2-13B, with rephrasing methods listed on the left and corresponding rephrases on the right.

*be the probability of sampling the most probable answer $A \in \mathcal{Y}$ under transformations $\mathcal{T}(\boldsymbol{x}) \sim \tau$. Let $\boldsymbol{z}_{mean} + \epsilon_{rephrase}$ be the latent representation of $\boldsymbol{x}$ under $\mathcal{T}(\boldsymbol{x})$ at the final LLM layer, where $\boldsymbol{z}_{mean}$ is the mean representation and $\epsilon_{rephrase}$ is some additive noise. Let $\mathbf{w}$ be the separating hyperplane between the most probable answer $A$ and the second most probable answer $B$. Assuming that $\mathbf{w}^\top \epsilon_{rephrase} \sim \rho$ follows a logistic distribution with $\mu = 0$ and $s = 1$ then*

$$p_A(\boldsymbol{x}) = p(A|\boldsymbol{z}_{mean}, f) \tag{6}$$

*where $p(A|\boldsymbol{z}_{mean}, f)$ is the probability of $A$ given $\boldsymbol{z}_{mean}$ under the categorical distribution of the final layer.*

*Proof.* We first analyze the categorical distribution, resulting from applying the softmax on the final layer logits. In the binary classification case given a top-1 class prediction $A$, the softmax probability of this class is

$$
\begin{aligned}
p(A|\boldsymbol{x}, f) &= \frac{e^{\mathbf{w}_A^\top \boldsymbol{z} + b_A}}{e^{\mathbf{w}_A^\top \boldsymbol{z} + b_A} + e^{\mathbf{w}_B^\top \boldsymbol{z} + b_B}} \\
&= \frac{1}{1 + e^{-(\mathbf{w}_A + b_A - \mathbf{w}_B - b_B)^\top \boldsymbol{z}}} = \frac{1}{1 + e^{-(\mathbf{w}^\top \boldsymbol{z} + b)}}.
\end{aligned}
\tag{7}
$$

The above simply corresponds to the folk knowledge that a softmax layer with two classes is equivalent to a single separating hyperplane that assigns classes based on the rule $\mathrm{sign}\left(\mathbf{w}^\top \boldsymbol{z} + b\right)$, specifically

$$
g(\boldsymbol{z}) = \begin{cases} A & \text{if } \left(\mathbf{w}^\top \boldsymbol{z} + b\right) > 0, \\ B & \text{otherwise.} \end{cases}
$$

After establishing that the softmax layer is equivalent to this single separating hyperplane, let us relate $p_A(\boldsymbol{x})$ to $\mathbf{w}^\top \boldsymbol{z} + b$. We have

$$
\begin{aligned}
p_A(\boldsymbol{x}) &= \mathbb{P}\left(f(\mathcal{T}(\boldsymbol{x})) = A\right) \\
&= \mathbb{P}\left(\mathbf{w}^\top (\boldsymbol{z}_{mean} + \epsilon_{rephrase}) + b > 0\right) \\
&= \mathbb{P}\left(\mathbf{w}^\top \boldsymbol{z}_{mean} + \mathbf{w}^\top \epsilon_{rephrase} + b > 0\right) \\
&= \mathbb{P}\left(Z > -\mathbf{w}^\top \boldsymbol{z}_{mean} - b\right) \\
&= 1 - \mathbb{P}\left(Z < -\mathbf{w}^\top \boldsymbol{z}_{mean} - b\right) \\
&= 1 - F\left(-\mathbf{w}^\top \boldsymbol{z}_{mean} - b\right)
\end{aligned}
\tag{8}
$$

Then $F(-\mathbf{w}^\top \mathbf{z}_{mean} - b) = 1 - p_A \iff \mathbf{w}^\top \mathbf{z}_{mean} + b = -F^{-1}(1 - p_A)$. We substitute this result to 7, assume that $F$ is the cumulative of the logistic distribution with $\mu = 0$ and $s = 1$ and get

$$p(A|\mathbf{z}_{mean}, f) = \frac{1}{1 + e^{F^{-1}(1-p_A)}} \tag{9}$$

$$= \frac{1}{1 + e^{-F^{-1}(p_A(\mathbf{x}))}} \tag{10}$$

$$= p_A(\mathbf{x}) \tag{11}$$

In the second line we used the fact that the inverse cumulative $F^{-1}$ of the logistic distribution is symmetric around 0.5. In the third line we use the fact that $\frac{1}{1+e^{-x}}$ is the cumulative of the logistic with $\mu = 0$ and $s = 1$. Thus $p(A|\mathbf{z}_{mean}, f) = F(F^{-1}(p_A(\mathbf{x}))) \iff p(A|\mathbf{z}_{mean}, f) = p_A(\mathbf{x})$

A technical point remains. Even though in the previous we can assume that $g(\mathbf{z}_{mean}) = A$ (that $\mathbf{z}_{mean}$ results in the most probable class) by definition, we still need to show that $A = \mathrm{argmax}_i \mathbb{P}\left(f(\mathcal{T}(\mathbf{x})) = i\right) \iff g(\mathbf{z}_{mean}) = A$. This means that for a closed-source LLM we can identify the (unknown) top-1 class A through Monte Carlo sampling ($A = \mathrm{argmax}_i \mathbb{P}\left(f(\mathcal{T}(\mathbf{x})) = i\right)$).

$$
\begin{aligned}
A = \mathrm{argmax}_i \mathbb{P}\left(f(\mathcal{T}(\mathbf{x})) = i\right) &\iff \mathbb{P}\left(f(\mathcal{T}(\mathbf{x})) = A\right) > \frac{1}{2} \\
&\iff \mathbb{P}\left(\mathbf{w}^\top (\mathbf{z}_{mean} + \epsilon_{rephrase}) + b \geq 0\right) > \frac{1}{2} \\
&\iff \mathbb{P}\left(\mathbf{w}^\top \mathbf{z}_{mean} + \mathbf{w}^\top \epsilon_{rephrase} + b \geq 0\right) > \frac{1}{2} \\
&\iff \mathbb{P}\left(Z \geq -\mathbf{w}^\top \mathbf{z}_{mean} - b\right) > \frac{1}{2} \\
&\iff \mathbb{P}\left(Z \leq \mathbf{w}^\top \mathbf{z}_{mean} + b\right) > \frac{1}{2} \\
&\iff \mathbf{w}^\top \mathbf{z}_{mean} + b > 0 \\
&\iff g(\mathbf{z}_{mean}) = A
\end{aligned} \tag{12}
$$

where we use the assumption that $Z$ follows a logistic distribution with $\mu = 0$ and $s = 1$. $\square$

**Proposition C.2.** *Let $g : \mathbb{R}^{d_\eta} \to \mathcal{Y}$ be the final encoder layer of an LLM, $\mathbf{x}$ is a base query and $\mathcal{T}(\mathbf{x}) \sim \tau$ is some randomized transformation of the base query. Let*

$$p_A(\mathbf{x}) = \mathbb{P}\left(f(\mathcal{T}(\mathbf{x})) = A\right), \tag{13}$$

*be the probability of sampling the most probable answer $f(\mathbf{x}) = A \in \mathcal{Y}$ under transformations $\mathcal{T}(\mathbf{x}) \sim \tau$. Let $\mathbf{z}_{mean} + \epsilon_{topk} + \epsilon_{rephrase}$ be the latent representation of $\mathbf{x}$ under $\mathcal{T}(\mathbf{x})$ at the final LLM layer, where $\mathbf{z}_{mean}$ is the mean representation and $\epsilon_{topk}$ is additive noise resulting from the top-k decoding and $\epsilon_{rephrase}$ is additive noise resulting from the rephrasings $\mathcal{T}(\mathbf{x})$. Assuming that $\mathbf{w}^\top (\epsilon_{topk} + \epsilon_{rephrase}) \sim \rho$ approximately follows a logistic distribution with $\mu = 0$ and $s = \sqrt{s_{topk}^2 + s_{rephrase}^2}$ then*

$$p_A(\mathbf{x}) \approx 0.5 + \frac{1}{\sqrt{s_{topk}^2 + s_{rephrase}^2}}(p(A|\mathbf{z}_{mean}, f) - 0.5) \tag{14}$$

*where $p(A|\mathbf{z}_{mean}, f)$ is the probability of A given $\mathbf{z}_{mean}$ under the categorical distribution of g.*

*Proof.* We first claim that the sum of two logistic distributions $(\mu_1, s_1)$ and $(\mu_1, s_1)$ is approximately logistic with $(\mu_1 + \mu_2, \sqrt{s_1^2 + s_2^2})$ by claiming that logistic distributions are approximately Gaussian. Then considering that $p(A|\mathbf{z}_{mean}, f) = \frac{1}{1+e^{F^{-1}(1-p_A(\mathbf{x}))}}$ we can write

$$
\begin{aligned}
p(A|\mathbf{z}_{mean}, f) &= \frac{1}{1 + e^{F^{-1}(1-p_A(\mathbf{x}))}} = \frac{1}{1 + e^{-F^{-1}(p_A(\mathbf{x}))}} \\
&= 0.5 + \frac{1}{4} F^{-1}(p_A(\mathbf{x})) = 0.5 + \frac{1}{4} 4\sqrt{s_{topk}^2 + s_{rephrase}^2}(p_A(\mathbf{x}) - 0.5)
\end{aligned} \tag{15}
$$

Table 7: Comparisons between our best rephrasing method and CoT. Our rephrasing method obtains comparable results to CoT in terms of Brier score and other calibration metrics.

| Dataset | Model | Method | Acc ↑ | ECE ↓ | TACE ↓ | Brier ↓ | AUROC ↑ |
|---|---|---|---|---|---|---|---|
| ARC-C | Mistral-7B | CoT | 0.725 | 0.173 | 0.071 | 0.439 | 0.719 |
| | | expansion | 0.602 | 0.133 | 0.099 | 0.509 | 0.847 |
| | Llama-2-7B | CoT | 0.407 | 0.205 | 0.151 | 0.783 | 0.696 |
| | | expansion | 0.373 | 0.112 | 0.153 | 0.778 | 0.687 |
| | Llama-2-13B | CoT | 0.369 | 0.137 | 0.148 | 0.782 | 0.729 |
| | | reword | 0.445 | 0.084 | 0.119 | 0.714 | 0.721 |
| ARC-E | Mistral-7B | CoT | 0.857 | 0.07 | 0.037 | 0.211 | 0.829 |
| | | reword | 0.753 | 0.045 | 0.062 | 0.297 | 0.931 |
| | Llama-2-7B | CoT | 0.482 | 0.104 | 0.116 | 0.624 | 0.842 |
| | | rephrase | 0.535 | 0.131 | 0.117 | 0.603 | 0.830 |
| | Llama-2-13B | CoT | 0.463 | 0.097 | 0.124 | 0.61 | 0.884 |
| | | expansion | 0.524 | 0.078 | 0.12 | 0.552 | 0.893 |
| OBQA | Mistral-7B | CoT | 0.662 | 0.153 | 0.083 | 0.501 | 0.762 |
| | | reword | 0.552 | 0.105 | 0.102 | 0.592 | 0.796 |
| | Llama-2-7B | CoT | 0.39 | 0.185 | 0.145 | 0.805 | 0.713 |
| | | expansion | 0.362 | 0.083 | 0.138 | 0.775 | 0.678 |
| | Llama-2-13B | CoT | 0.37 | 0.166 | 0.153 | 0.801 | 0.683 |
| | | rephrase | 0.428 | 0.095 | 0.14 | 0.729 | 0.73 |

In the first line we first considered that $F^{-1}$ for the logistic is symmetric thus $F^{-1}(1 - p_A(\boldsymbol{x})) = -F^{-1}(p_A(\boldsymbol{x}))$. In the second line we first do a first order Taylor expansion around 0 on $\frac{1}{1+e^{-x}}$ and then a first order Taylor expansion around 0.5 on $F^{-1}$. □

# D    Additional comparisons with CoT

We compare with Chain-of-Thought Wei et al. (2022) for uncertainty estimation and plot the results in Table 7. We find that we get competitive results with CoT. At the same time our method is significantly easier and more natural to implement for humans interacting via text with an LLM. In CoT one needs to first obtain a sequence of reasoning steps. These should then be used as additional context when asking an LLM to answer again the base question. By contrast we propose a simple one step process of rephrasing the base question.

# E    Additional results

We used three different models, the Llama-2 7B model, the Llama-2 13B model (Touvron et al., 2023) and the Mistral 7B model (Jiang et al., 2023). We tested our framework on three multiple choice tasks of different difficulty namely ARC-Challenge, ARC-Easy (Clark et al., 2018), and Openbookqa (Mihaylov et al., 2018). Following Kojima et al. (2022), we extract the answer from LLM-generated texts by looking at the first appearance of A/B/C/D. To test for calibration we used standard calibration metrics, including the ECE and TACE (Naeini et al., 2015), Brier score (Murphy, 1973) and AUROC (Murphy, 2012). We note that for a fair comparison when the accuracy drops significantly, we must consult the Brier score which is a proper scoring rule. This is because, the ECE, TACE and AUROC are not proper scoring rules and can in general trade-off accuracy for calibration. For a baseline, we assumed 100% confidence for each deterministic prediction. We also tested the "hint" based approach of Xiong et al. (2023), which we describe in detail in Appendix A.

We present the results in Tables 11, 12 and 13. In the majority of cases rephrasing outperforms the naive baseline by $10 - 40\%$ in AUROC, $10 - 30\%$ in ECE, and $0 - 0.4$ in Brier. Our approach also typically outperforms the "hint" base approach of Xiong et al. (2023) by $10 - 20\%$ in AUROC, $5 - 10\%$ in ECE, and $0.1$ in Brier. In particular, the "hint" based approach which more inflexible than our approach and typically hurts accuracy significantly $10 - 20\%$ compared to $5 - 10\%$ for our

Table 8: Evaluation results on ARC-Challenge with various rephrasing methods applied to three LLMs. In the majority of cases, the rephrasing approach outperforms the naive baseline by $10 - 40\%$ in AUROC, $10 - 30\%$ in ECE and $0 - 0.4$ in Brier.

| Model | Rephrasing | Acc ↑ | ECE ↓ | TACE ↓ | Brier ↓ | AUROC ↑ | temp |
|---|---|---|---|---|---|---|---|
| Mistral-7B | top-1 | 0.742 | 0.258 | **0.065** | 0.517 | 0.5 | - |
| | hint | 0.593, | 0.201, | 0.108, | 0.614, | 0.695, | - |
| | reword | 0.619 | 0.12 | 0.103 | 0.512 | **0.846** | 1.0 |
| | rephrase | 0.555 | 0.125 | 0.103 | 0.571 | 0.817 | 1.5 |
| | paraphrase | 0.525 | **0.102** | 0.115 | 0.592 | 0.827 | 1.5 |
| | expansion | 0.602 | 0.133 | 0.099 | **0.509** | 0.847 | 1.0 |
| Llama-2-7B | top-1 | 0.483 | 0.517 | - | 1.034 | 0.5 | - |
| | hint | 0.258, | **0.071**, | **0.144**, | 0.839, | 0.562, | - |
| | reword | 0.352 | 0.193 | 0.176 | 0.853 | 0.626 | 1.5 |
| | rephrase | 0.381 | 0.263 | 0.173 | 0.871 | 0.656 | 1.5 |
| | paraphrase | 0.39 | 0.287 | 0.162 | 0.883 | 0.67 | 1.0 |
| | expansion | 0.373 | 0.112 | 0.153 | **0.778** | **0.687** | 1.5 |
| Llama-2-13B | top-1 | 0.508 | 0.492 | - | 0.983 | 0.5 | - |
| | hint | 0.331, | 0.147, | 0.134, | 0.813, | 0.57, | - |
| | reword | 0.445 | **0.084** | **0.119** | **0.714** | 0.721 | 1.5 |
| | rephrase | 0.441 | 0.128 | 0.134 | 0.727 | 0.713 | 1.5 |
| | paraphrase | 0.453 | 0.092 | 0.129 | 0.717 | 0.697 | 1.5 |
| | expansion | 0.441 | 0.154 | 0.142 | **0.715** | **0.784** | 1.2 |

approach. For our method, these accuracy drops are more prevalent in the smaller 7B models, while the larger 13B model often shows a much smaller drop.

Crucially, the different rephrasing methods exhibit different calibration gains. On average, in terms of all calibration metrics the best methods are the "expansion" and "reword" methods, which make the queries more verbose, and substitute words with synonyms respectively. In terms of AUROC "expansion" outperforms the alternatives by $1 - 5\%$. In terms of the Brier score it outperforms by $\approx 0.05$. To instantiate our rephrasings we used a prompt with a one-shot example and a temperature parameter resulting in greater or smaller varieties of rephrasings. We include this temperature parameter in the Tables. Generally, we choose this temperature that balances accuracy and calibration. In Figure 4 we plot the behaviour as the number of MC draws increases.

In Appendix D, we also compare with Chain-of-Thought Wei et al. (2022) for uncertainty estimation. We find that we get competitive results with CoT. At the same time our method is significantly easier and more natural to implement for humans interacting via text with an LLM.

In Tables 11, 12 and 13 and Figure 3, we present the results for the top-k experiments with and without rephrasing, with $k = 40$. We also present the relaxed temperature sampling variant Wei et al. (2022). We see that the stochasticity of top-40 compared to top-1 decoding from Tables 8, 9 and 10 *results in an improvement in calibration but a drop in accuracy. The Brier score often improves at the cost of accuracy*. Further stochasticity in answers caused by rephrasings has a similar effect. These observations are consistent with the fact that top-k and nucleus sampling (Holtzman et al., 2019) make text more human-like but not necessarily more "accurate".

Table 9: Evaluation results on ARC-Easy with various rephrasing methods applied to three LLMs. In the majority of cases, the rephrasing approach outperforms the naive baseline by $10-40\%$ in AUROC, $10-30\%$ in ECE, and $0-0.4$ in Brier.

| Model | Rephrasing | Acc ↑ | ECE ↓ | TACE ↓ | Brier ↓ | AUROC ↑ | temp |
|---|---|---|---|---|---|---|---|
| | top-1 | 0.866 | 0.134 | **0.034** | **0.269** | 0.5 | - |
| | hint | 0.773, | 0.17, | 0.076, | 0.386, | 0.795, | - |
| Mistral-7B | reword | 0.753 | 0.045 | 0.062 | 0.297 | 0.931 | 1.0 |
| | rephrase | 0.678 | 0.035 | 0.076 | 0.357 | **0.953** | 1.5 |
| | paraphrase | 0.663 | 0.036 | 0.08 | 0.381 | 0.943 | 1.5 |
| | expansion | 0.742 | **0.034** | 0.067 | 0.31 | 0.936 | 1.0 |
| | top-1 | 0.672 | 0.328 | 0.082 | 0.656 | 0.5 | - |
| | hint | 0.231, | **0.041**, | 0.149, | 0.827, | 0.663, | - |
| Llama-2-7B | reword | 0.43 | 0.084 | **0.119** | 0.672 | 0.818 | 1.5 |
| | rephrase | 0.535 | 0.131 | **0.117** | **0.603** | 0.830 | 1.5 |
| | paraphrase | 0.526 | 0.184 | 0.125 | 0.626 | **0.831** | 1.0 |
| | expansion | 0.405 | 0.045 | **0.119** | 0.692 | 0.818 | 1.5 |
| | top-1 | 0.617 | 0.383 | **0.096** | 0.767 | 0.5 | - |
| | hint | 0.346, | **0.089**, | 0.128, | 0.77, | 0.673, | - |
| Llama-2-13B | reword | 0.546 | 0.07 | 0.11 | 0.58 | 0.814 | 1.5 |
| | rephrase | 0.526 | 0.07 | 0.112 | 0.579 | 0.842 | 1.5 |
| | paraphrase | 0.518 | 0.104 | 0.119 | 0.604 | 0.815 | 1.5 |
| | expansion | 0.524 | 0.078 | 0.12 | **0.552** | **0.893** | 1.2 |

Table 10: Evaluation results on OpenBookQA with various rephrasing methods applied to three LLMs. In the majority of cases, the rephrasing approach outperforms the naive baseline by $10-40\%$ in AUROC, $10-30\%$ in ECE, and $0-0.4$ in Brier.

| Model | Rephrasing | Acc ↑ | ECE ↓ | TACE ↓ | Brier ↓ | AUROC ↑ | temp |
|---|---|---|---|---|---|---|---|
| | top-1 | 0.655 | 0.345 | **0.086** | 0.69 | 0.5 | - |
| | hint | 0.56, | 0.265, | 0.119, | 0.71, | 0.606, | - |
| Mistral-7B | reword | 0.552 | 0.105 | 0.102 | **0.592** | 0.796 | 1.0 |
| | rephrase | 0.482 | 0.107 | 0.122 | 0.641 | 0.809 | 1.5 |
| | paraphrase | 0.49 | **0.076** | 0.116 | 0.622 | 0.826 | 1.5 |
| | expansion | 0.518 | 0.087 | 0.117 | 0.596 | **0.837** | 1.0 |
| | top-1 | 0.478 | 0.522 | **0.131** | 1.045 | 0.5 | - |
| | hint | 0.275, | **0.08**, | 0.142, | 0.832, | 0.556, | - |
| Llama-2-7B | reword | 0.388 | 0.137 | 0.143 | 0.786 | 0.689 | 1.5 |
| | rephrase | 0.39 | 0.196 | 0.156 | 0.806 | **0.721** | 1.5 |
| | paraphrase | 0.398 | 0.227 | 0.159 | 0.834 | 0.712 | 1.0 |
| | expansion | 0.362 | 0.083 | 0.138 | **0.775** | 0.678 | 1.5 |
| | top-1 | 0.418 | 0.582 | - | 1.165 | 0.5 | - |
| | hint | 0.295, | **0.069**, | **0.138**, | 0.809, | 0.613, | - |
| Llama-2-13B | reword | 0.428 | 0.117 | 0.142 | 0.75 | 0.676 | 1.5 |
| | rephrase | 0.428 | 0.095 | 0.14 | **0.729** | 0.73 | 1.5 |
| | paraphrase | 0.41 | 0.116 | 0.141 | 0.759 | 0.682 | 1.5 |
| | expansion | 0.41 | 0.143 | 0.147 | 0.772 | **0.702** | 1.2 |

Table 11: Evaluation results on ARC-Challenge with various rephrasing methods applied to three LLMs using top-k decoding. In the majority of cases rephrasing + top-k outperforms simple top-k in terms of calibration.

| Model | Rephrasing | Acc ↑ | ECE ↓ | TACE ↓ | Brier ↓ | AUROC ↑ | temp |
|---|---|---|---|---|---|---|---|
| Mistral-7B | top-k | 0.746, | 0.272, | 0.091, | 0.511, | 0.6, | - |
| | temp-sampling | 0.742 | 0.272 | 0.089 | 0.513 | 0.605 | - |
| | reword | 0.547, | **0.05**, | 0.093, | 0.543, | **0.864**, | 1.5 |
| | rephrase | 0.64, | 0.106, | **0.086**, | **0.485**, | 0.82, | 1.0 |
| | paraphrase | 0.631, | 0.11, | 0.098, | 0.495, | 0.83, | 1.0 |
| | expansion | 0.517, | 0.061, | 0.114, | 0.573, | 0.859, | 1.5 |
| Llama-2-7B | top-k | 0.436, | 0.201, | **0.139**, | **0.761**, | 0.602, | - |
| | temp-sampling | 0.441 | 0.211 | 0.132 | 0.757 | 0.621 | - |
| | reword | 0.335, | 0.187, | 0.166, | 0.858, | 0.62, | 1.5 |
| | rephrase | 0.356, | 0.314, | 0.17, | 0.944, | 0.627, | 1.0 |
| | paraphrase | 0.309, | 0.185, | 0.162, | 0.851, | **0.69**, | 1.5 |
| | expansion | 0.322, | **0.144**, | 0.155, | 0.828, | 0.622, | 1.5 |
| Llama-2-13B | top-k | 0.462, | 0.125, | **0.115**, | **0.679**, | **0.753**, | - |
| | temp-sampling | 0.47, | 0.122 | 0.115 | 0.662 | 0.766 | - |
| | reword | 0.352, | 0.087, | 0.136, | 0.771, | 0.687, | 1.5 |
| | rephrase | 0.398, | **0.068**, | 0.136, | 0.725, | 0.743, | 1.0 |
| | paraphrase | 0.364, | 0.109, | 0.137, | 0.738, | 0.719, | 1.2 |
| | expansion | 0.373, | 0.124, | 0.143, | 0.76, | 0.669, | 1.5 |

Table 12: Evaluation results on ARC-Easy with various rephrasing methods applied to three LLMs using top-k decoding. In the majority of cases rephrasing + top-k outperforms simple top-k in terms of calibration.

| Model | Rephrasing | Acc ↑ | ECE ↓ | TACE ↓ | Brier ↓ | AUROC ↑ | temp |
|---|---|---|---|---|---|---|---|
| Mistral-7B | top-k | 0.868, | 0.133, | **0.042**, | **0.255**, | 0.695, | - |
| | temp-sampling | 0.859 | 0.131 | 0.046 | 0.266 | 0.677 | - |
| | reword | 0.694, | 0.054, | 0.076, | 0.344, | 0.941, | 1.5 |
| | rephrase | 0.789, | 0.047, | 0.049, | 0.274, | 0.911, | 1.0 |
| | paraphrase | 0.753, | **0.036**, | 0.056, | 0.3, | 0.922, | 1.0 |
| | expansion | 0.63, | 0.042, | 0.086, | 0.403, | **0.942**, | 1.5 |
| Llama-2-7B | top-k | 0.612, | 0.25, | 0.115, | 0.612, | 0.73, | - |
| | temp-sampling | 0.619 | 0.261 | 0.114 | 0.617 | 0.717 | - |
| | reword | 0.401, | **0.074**, | 0.121, | 0.681, | 0.825, | 1.5 |
| | rephrase | 0.564, | 0.145, | **0.108**, | **0.584**, | 0.819, | 1.0 |
| | paraphrase | 0.425, | 0.08, | 0.117, | 0.665, | **0.835**, | 1.5 |
| | expansion | 0.335, | 0.054, | 0.138, | 0.742, | 0.791, | 1.5 |
| Llama-2-13B | top-k | 0.557, | 0.06, | **0.098**, | **0.528**, | **0.865**, | - |
| | temp-sampling | 0.544 | 0.087 | 0.107 | 0.532 | 0.866 | - |
| | reword | 0.412, | 0.106, | 0.129, | 0.72, | 0.741, | 1.5 |
| | rephrase | 0.458, | **0.05**, | 0.12, | 0.643, | 0.817, | 1.0 |
| | paraphrase | 0.427, | 0.066, | 0.126, | 0.652, | 0.845, | 1.2 |
| | expansion | 0.366, | 0.087, | 0.13, | 0.74, | 0.75, | 1.5 |

Table 13: Evaluation results on OpenBookQA with various rephrasing methods applied to three LLMs using top-k decoding. In the majority of cases rephrasing + top-k outperforms simple top-k in terms of calibration.

| Model | Rephrasing | Acc ↑ | ECE ↓ | TACE ↓ | Brier ↓ | AUROC ↑ | temp |
|---|---|---|---|---|---|---|---|
| Mistral-7B | top-k | 0.638, | 0.289, | 0.101, | 0.636, | 0.636, | - |
| | temp-sampling | 0.668 | 0.289 | 0.098 | 0.607 | 0.624 | - |
| | reword | 0.528, | 0.103, | 0.105, | 0.606, | 0.794, | 1.5 |
| | rephrase | 0.582, | 0.109, | **0.093**, | **0.542**, | **0.821**, | 1.0 |
| | paraphrase | 0.552, | 0.078, | 0.101, | 0.57, | 0.817, | 1.0 |
| | expansion | 0.445, | **0.061**, | 0.128, | 0.653, | 0.818, | 1.5 |
| Llama-2-7B | top-k | 0.412, | 0.208, | **0.129**, | **0.776**, | 0.617, | - |
| | temp-sampling | 0.442 | 0.235 | 0.13 | 0.772 | 0.599 | - |
| | reword | 0.34, | 0.14, | 0.153, | 0.807, | 0.696, | 1.5 |
| | rephrase | 0.408, | 0.239, | 0.154, | 0.815, | 0.704, | 1.0 |
| | paraphrase | 0.355, | 0.127, | 0.145, | 0.783, | **0.721**, | 1.5 |
| | expansion | 0.308, | **0.098**, | 0.151, | 0.807, | 0.711, | 1.5 |
| Llama-2-13B | top-k | 0.43, | 0.114, | **0.13**, | **0.708**, | **0.72**, | - |
| | temp-sampling | 0.43, | 0.099 | 0.121 | 0.702 | 0.733 | - |
| | reword | 0.345, | 0.111, | 0.144, | 0.794, | 0.618, | 1.5 |
| | rephrase | 0.345, | **0.062**, | 0.141, | 0.767, | 0.706, | 1.0 |
| | paraphrase | 0.37, | 0.092, | 0.141, | 0.763, | 0.67, | 1.2 |
| | expansion | 0.36, | 0.138, | 0.138, | 0.799, | 0.574, | 1.5 |

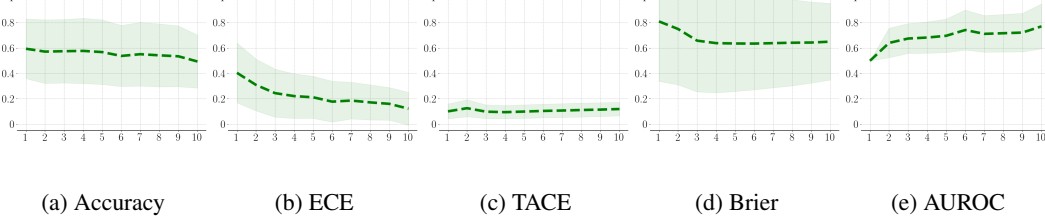

(a) Accuracy     (b) ECE     (c) TACE     (d) Brier     (e) AUROC

Figure 4: The behavior of the Accuracy, ECE, TACE, Brier, and AUROC for all datasets, architectures, and expansion methods, as we increase the number of samples. We plot the average value as well as confidence intervals $\pm 2\sigma$. We see that the ECE and the AUROC improve with more samples while the accuracy drops slightly. This might be because the meaning of some queries is completely destroyed by our rephrasings. The Brier score captures this tradeoff by having a minimum at approximately 5 samples. The TACE remains relatively stable with respect to the number of samples.

