# OpenReview forum: "Just rephrase it! Uncertainty estimation in closed-source language models via multiple rephrased queries"
_NeurIPS.cc/2024/Workshop/SafeGenAi — SafeGenAi Poster_

### Official Review · Reviewer_Ryau · 2024-10-09
**Adequate Experiments and Theoretical Inferences.**

**Rating:** 7
**Confidence:** 3

**Review:**

This article primarily focuses on estimating the uncertainty of closed-source LLMs via multiple rephrasings of an original base query. It introduces a simple yet practical rephrasing method and proposes a theoretical framework to illustrate why multiple rephrased queries obtain calibrated uncertainty estimates. Additionally, the paper designs several experiments to validate this approach and its theoretical framwork across multiple datasets and settings.

---

### Official Review · Reviewer_LPEY · 2024-10-09
**Practical method, partially good empirical results, but the Proposition is still improvable**

**Rating:** 6
**Confidence:** 3

**Review:**

This paper presents a simple and practical way to measure the model uncertainty when answering multiple choice questions. The authors provide various metrics to show that their proposed method can perform better than other baselines. They also provide some intriguing theoretical analysis of the method. Despite these pros, there are the following concerns:

1. The assumption in Proposition C.2 that $\operatorname{Log}(\mu_1+\mu_2, \sqrt{s_1^2+s_2^2})$
 is approximately Gaussian may be too bold. In fact, when $s_1$ and $s_2$ gets larger, the tails of the logistic can be much larger than Gaussian. I am not sure in that case how useful this Proposition is.

2. The authors mainly experiment with llama2-7b, 13b and Mistral-7b, and there are nice results on these models. However, by the time there are some more capable LLMs of similar sizes available (e.g. llama3-8b.) It would be better to include results from these more recent models, or an explanation of why not. Was it because they have better representations and logits that the proposed method cannot significantly outperform, or was it just the experiments have not been conducted?

---

### Official Review · Reviewer_89M7 · 2024-10-09
**Review for the Paper: "Just rephrase it! Uncertainty estimation in closed-source language models via multiple rephrased queries"**

**Rating:** 8
**Confidence:** 4

**Review:**

### **Summary**

This paper introduces a novel method for estimating uncertainty in language models (LLMs) through multiple rephrased queries of a base prompt. The authors argue that by rephrasing questions and analyzing the consistency of responses, it is possible to better estimate the model’s uncertainty. Their method aims to improve upon existing techniques by providing easily memorizable rephrasing rules and a theoretical framework to justify the calibration improvements observed with these strategies. The paper evaluates various rephrasing methods (e.g., rewording, rephrasing, paraphrasing, expanding) across different datasets and models (such as Llama-2 and Mistral) and shows that this approach can achieve uncertainty estimates comparable to methods that have direct access to model logits.

### **Strengths**

1. **Practical Rephrasing Rules**: The proposed method uses intuitive and easy-to-apply rephrasing strategies, making it accessible even for users without advanced technical knowledge
2. **Improvement in Calibration**: The paper demonstrates significant improvements in uncertainty calibration across multiple models and datasets, outperforming baseline methods
3. **Comparison to Logit-based Methods**: Despite not having access to model logits, the rephrasing methods achieve results comparable to logit-based uncertainty estimation, which is typically only available in open-source models

### **Weaknesses**

1. **Limited Scope of Application**: While the method performs well in multiple-choice tasks, it may be less effective or harder to implement for more complex open-ended tasks that do not have clear-cut answers
2. **Empirical Validation Focuses on Specific Models**: The experiments are limited to a few models (Llama-2 and Mistral), and broader applicability to other architectures (like GPT models) or use cases might need further exploration
3. **Accuracy Trade-off**: In some cases, calibration improvements come at the cost of accuracy, particularly in smaller models where the method introduces a noticeable performance drop

### **Detailed Comments**

- The paper’s results show consistent gains in calibration metrics such as AUROC and Brier score. For instance, in the ARC-Challenge dataset, rephrasing methods showed up to a 40% improvement over the baseline in AUROC. However, the paper could have benefited from additional exploration into how well these results generalize to other types of models or different forms of language generation tasks
- The inclusion of formal propositions and proofs (e.g., Proposition 3.1 and 4.1) adds depth to the paper’s claims. These sections explain why rephrasing can temper the uncertainty of top-k decoding and how the rephrasing noise contributes to more calibrated predictions